# Gut Microbiota Mediates the Protective Effects of Traditional Chinese Medicine Formula Qiong-Yu-Gao against Cisplatin-Induced Acute Kidney Injury

Ye-Ting Zou,[a,b] Jing Zhou,[a] Jin-Hao Zhu,[a,b] Cheng-Ying Wu,[a] Hong Shen,[b] Wei Zhang,[a] Shan-Shan Zhou,[b] Jin-Di Xu,[a] Qian Mao,[b] Ye-Qing Zhang,[c] Fang Long,[a,c] Song-Lin Li[a,b]

[a]Department of Pharmaceutical Analysis, Affiliated Hospital of Integrated Traditional Chinese and Western Medicine, Nanjing University of Chinese Medicine, Nanjing, China
[b]Department of Metabolomics, Jiangsu Province Academy of Traditional Chinese Medicine, Nanjing, China
[c]Department of Respiratory Medicine, Affiliated Hospital of Integrated Traditional Chinese and Western Medicine, Nanjing University of Chinese Medicine, Nanjing, China

**ABSTRACT** Our previous study found that Qiong-Yu-Gao (QYG), a traditional Chinese medicine formula derived from Rehmanniae Radix, Poria, and Ginseng Radix, has protective effects against cisplatin-induced acute kidney injury (AKI), but the underlying mechanisms remain unknown. In the present study, the potential role of gut microbiota in the nephroprotective effects of QYG was investigated. We found that QYG treatment significantly attenuated cisplatin-induced AKI and gut dysbiosis, altered the levels of bacterial metabolites, with short-chain fatty acids (SCFAs) such as acetic acid and butyric acid increasing and uremic toxins such as indoxyl sulfate and *p*-cresyl sulfate reducing, and suppressed histone deacetylase expression and activity. Spearman's correlation analysis found that QYG-enriched fecal bacterial genera *Akkermansia*, *Faecalibaculum*, *Bifidobacterium*, and *Lachnospiraceae_NK4A136_group* were correlated with the altered metabolites, and these metabolites were also correlated with the biomarkers of AKI, as well as the indicators of fibrosis and inflammation. The essential role of gut microbiota was further verified by both the diminished protective effects with antibiotics-induced gut microbiota depletion and the transferable renal protection with fecal microbiota transplantation. All these results suggested that gut microbiota mediates the nephroprotective effects of QYG against cisplatin-induced AKI, potentially via increasing the production of SCFAs, thus suppressing histone deacetylase expression and activity, and reducing the accumulation of uremic toxins, thereby alleviating fibrosis, inflammation, and apoptosis in renal tissue.

**IMPORTANCE** Cisplatin-induced acute kidney injury is the main limiting factor restricting cisplatin's clinical application. Accumulating evidence indicated the important role of gut microbiota in pathogenesis of acute kidney injury. In the present study, we have demonstrated that gut microbiota mediates the protective effects of traditional Chinese medicine formula Qiong-Yu-Gao against cisplatin-induced acute kidney injury. The outputs of this study would provide scientific basis for future clinical applications of QYG as prebiotics to treat cisplatin-induced acute kidney injury, and gut microbiota may be a promising therapeutic target for chemotherapy-induced nephrotoxicity.

**KEYWORDS** gut microbiota, Qiong-Yu-Gao, acute kidney injury, cisplatin, short-chain fatty acids, uremic toxins, antibiotics-induced gut microbiota depletion, fecal microbiota transplantation

**Ad Hoc Peer Reviewer** zhihui zhou, Zhejiang University

Address correspondence to Song-Lin Li, songlinli64@126.com, or Fang Long, longfangcpu@163.com.

The authors declare no conflict of interest.

Acute kidney injury (AKI) is a common clinical syndrome characterized by the abrupt accumulation of end products of nitrogen metabolism (urea and creatinine), decreased urine output, or both (1). The acute changes in kidney function often lead

to chronic kidney disease, end-stage renal disease, and even death (2, 3). Nephrotoxic drugs are one of the major causes of AKI (4). Cisplatin is widely used in treatment of solid tumors (5). However, about 30% of cisplatin-administered patients suffered from renal dysfunction and injury, especially AKI (6). To date, there are few strategies for preventing cisplatin-induced AKI (7), and it is urgent to explore novel therapeutic measures to protect against nephrotoxicity of cisplatin.

Accumulating evidence suggests that gut microbiota is involved in the pathogenesis of AKI (8–10). In patients and animals with impaired renal function, the composition of gut microbiota is significantly altered (11). The gut microbiota dysbiosis may induce a predominant proteolytic fermentation and inflate the production of uremic toxins, such as indoxyl sulfate and $p$-cresyl sulfate (12, 13). Uremic toxins are known to accumulate in renal tissue and contribute to pathogenesis and disease progression by inducing fibrosis, inflammation, and apoptosis (14, 15). Meanwhile, short-chain fatty acids (SCFAs) are the end products of dietary fiber fermentation (16). As natural histone deacetylase (HDAC) inhibitors and substrates for metabolic reactions, SCFAs can attenuate kidney injury by regulating fibrosis, inflammation, apoptosis, and host metabolism (17–19). Pretreatment with probiotics is reported to slow the progression of acute and chronic kidney disease through increasing SCFA production and reducing the levels of uremic toxins (20). An orally delivered microbial cocktail has been shown to reduce urea and creatinine concentrations in animal models of AKI (21). These studies indicate that gut microbiota may be a potential therapeutic target for cisplatin-induced AKI.

Qiong-Yu-Gao (QYG), consisting of Rehmanniae Radix, Poria, and Ginseng Radix, is a commonly used tonic herbal formula that was first mentioned in the ancient Chinese medicine book named "Hong-Shi-Ji-Yan-Fang." Our previous work found that QYG has nephroprotective effects against cisplatin-induced AKI (22). However, the underlying mechanisms have not been well elucidated yet. Ever-increasing evidence suggests that polysaccharides and glycosides of herbal medicines are potential prebiotics (23, 24). QYG is rich in polysaccharides and glycosides (25, 26), so it is hypothesized that the nephroprotective effects of QYG against cisplatin-induced AKI might be mediated through remodeling the gut microbiota.

The present study aimed to explore the role of gut microbiota in nephroprotective effects of QYG in a mouse model of cisplatin-induced AKI. The nephroprotective effects of QYG were evaluated in terms of histological structure of kidney, biomarkers of AKI, and indicators of fibrosis, inflammation, and apoptosis. The diversity and function of gut microbiota were characterized using 16S rRNA gene sequencing. The related metabolites were analyzed by gas chromatography-mass spectrometry (GC-MS) and ultraperformance liquid chromatography with quadrupole time-of-flight tandem mass spectrometry (UPLC-QTOF-MS/MS)-based metabolomics analysis. The gut microbiota-dependent effects of QYG were further verified by antibiotic treatment and fecal microbiota transplantation (FMT).

## RESULTS

**QYG attenuated cisplatin-induced AKI by exerting antifibrotic, anti-inflammatory, and anti-apoptotic effects.** The nephroprotective effects of QYG were evaluated in a mouse model of cisplatin-induced AKI. Renal tubular injury, renal dysfunction, and body weight loss are commonly witnessed in AKI animal models. Tubular injury (tubular degeneration, swelling, vacuole formation, and necrosis) induced by cisplatin was mitigated by QYG treatment as accessed by hematoxylin and eosin (H&E) staining of the renal tissue (Fig. 1a). Compared with the cisplatin group (Cis group), QYG treatment significantly decreased blood urea nitrogen (BUN) in serum and suppressed the gene expression of kidney injury markers including hepatitis A virus cellular receptor 1 (HAVCR1) and lipocalin 2 (LCN2) in renal tissue ($P < 0.001$) (Fig. 1b and c). In addition, QYG increased the urine output and alleviated the body weight loss induced by cisplatin ($P < 0.01$) (Fig. 1d and e).

The results of Masson's trichrome staining showed that QYG alleviated the fibrosis

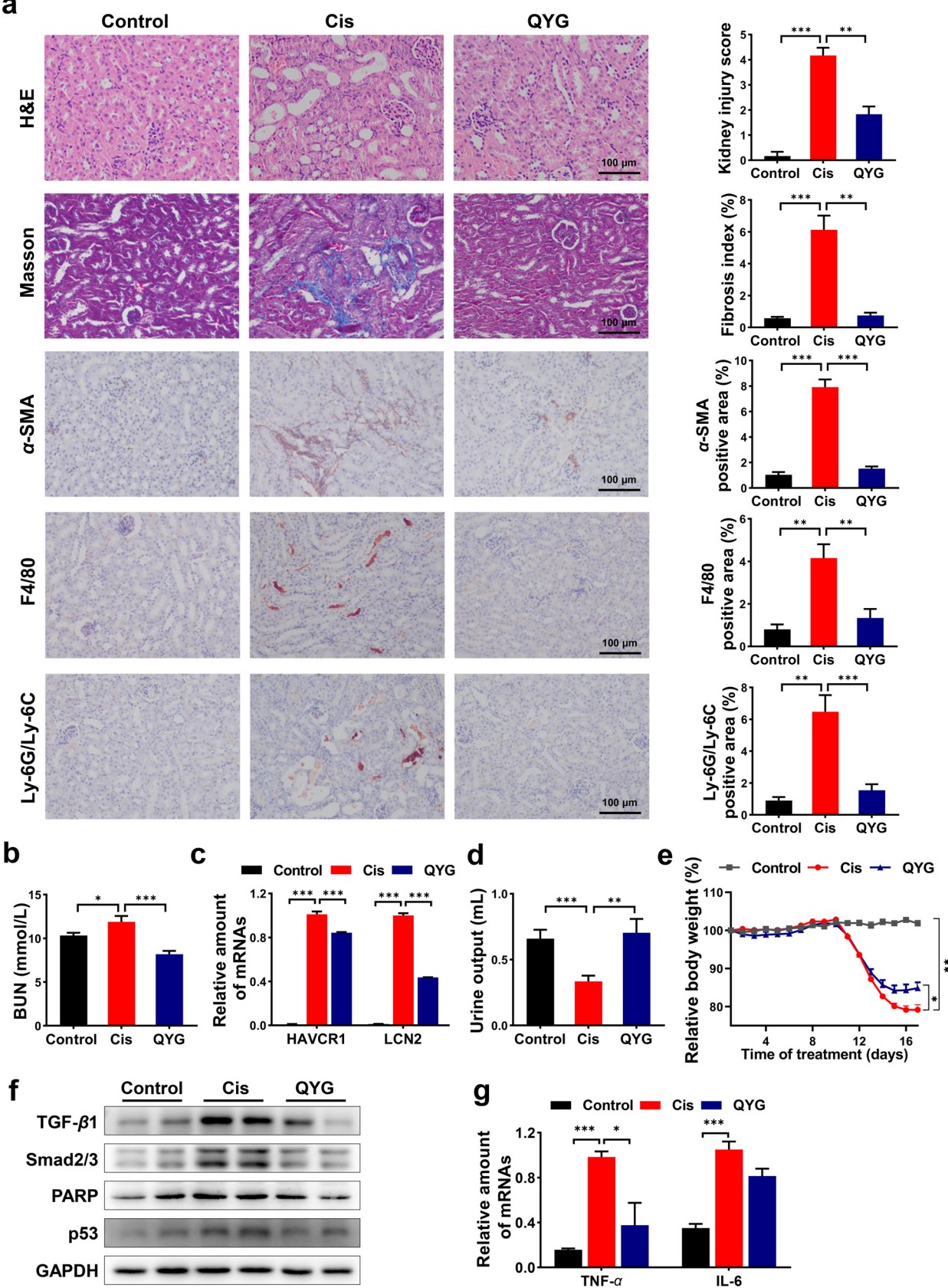

**FIG 1** QYG attenuated cisplatin-induced AKI, improved renal function, and inhibited potential mediators of cisplatin-induced nephrotoxicity. (a) Kidney obtained from treated mice was examined with H&E staining, Masson's trichrome staining, and IHC against α-SMA, F4/80, and Ly-6G/Ly-

**TABLE 1** Alpha-diversity of bacterial communities in fecal samples[a]

| | Shannon index for: | | | Invsimpson index for: | | |
|---|---|---|---|---|---|---|
| Group | OTU | Genus | Order | OTU | Genus | Order |
| Control | 3.41 ± 0.09 | 2.14 ± 0.07 | 1.59 ± 0.08 | 12.78 ± 1.59 | 5.00 ± 0.35 | 3.53 ± 0.36 |
| Cis | 2.89 ± 0.34# | 1.89 ± 0.08# | 1.38 ± 0.09 | 5.95 ± 0.66## | 3.93 ± 0.33 | 2.89 ± 0.26 |
| QYG | 3.03 ± 0.08** | 2.14 ± 0.06** | 1.72 ± 0.03** | 10.14 ± 1.19** | 5.07 ± 0.45** | 3.83 ± 0.32* |

[a]Data are presented as mean ± SEM ($n = 6$). #, $P < 0.05$, ##, $P < 0.01$, in comparison with control group; *, $P < 0.05$, **, $P < 0.01$, in comparison with Cis group.

induced by cisplatin (Fig. 1a). Transforming growth factor $\beta$ (TGF-$\beta$)/Smad signaling is a key pathway in renal fibrosis. Immunohistochemistry (IHC) staining and Western blotting showed that cisplatin significantly elevated the protein expression of $\alpha$-smooth muscle actin ($\alpha$-SMA), TGF-$\beta$1, and Smad2/3 (Fig. 1a and f), whereas QYG significantly decreased the production of these proteins. IHC staining showed that cisplatin significantly upregulated the expression of inflammation markers F4/80 and Ly-6G/Ly-6C (Fig. 1a), while QYG reduced the protein levels of these markers. QYG also suppressed the aberrant expression of inflammatory cytokines, including tumor necrosis factor $\alpha$ (TNF-$\alpha$) and interleukin 6 (IL-6) (Fig. 1g). In addition, QYG mitigated cisplatin-induced apoptosis by decreasing the protein levels of poly (ADP-ribose) polymerase (PARP) and p53 in kidney (Fig. 1f). These results suggested that QYG attenuated cisplatin-induced AKI through exerting antifibrotic, anti-inflammatory, and anti-apoptotic effects.

**QYG alleviated cisplatin-induced dysbiosis of gut microbiota.** It is acknowledged that composition and function of gut microbiota were closely related with pathogenesis of AKI. Hence, fecal samples were collected and examined by 16S rRNA gene sequencing. Shannon index and Invsimpson index are indicators positively correlated with gut microbiota diversity. The results of 16S rRNA gene sequencing showed that both Shannon index and Invsimpson index were decreased in Cis group (Table 1), and pretreatment of QYG significantly increased the diversity of bacteria. A distinct clustering of operational taxonomic unit (OTU) profiles was observed for control, Cis, and QYG groups (Fig. 2a). Variance analysis was further used to identify the specific bacterial phylotypes that were altered by QYG treatment. Sixty-one OTUs responsible for group discrimination between control and Cis groups were identified, of which 35 were decreased and 26 were increased in the Cis group (Fig. S1). Meanwhile, QYG treatment remarkably regulated 25 of these OTUs. Bacterial genera significantly different between groups were analyzed (Fig. 2b). *Akkermansia* and *Faecalibaculum* in QYG group were both increased more than 4-fold and *Ileibacterium* was decreased more than 12-fold compared to the Cis group. *Lachnospiraceae NK4A136 group* and *Coriobacteriaceae_UCG_002*, which decreased by cisplatin, showed more relative abundance in QYG group. QYG treatment also enriched SCFA-producing bacteria *Bifidobacterium* significantly. Compared with that in the control group, the relative abundance of *Allobaculum*, *Lactobacillus*, *Alloprevotella*, *Bacteroides*, and *Eubacterium_fissicatena_group* was higher in the Cis group, whereas QYG treatment was found to decrease the relative abundance of these genera.

To characterize the functional alterations of gut microbiota induced by cisplatin, functional profiles of microbial communities were predicted using phylogenetic investigation of communities by reconstruction of unobserved states (PICRUSt) based on 16S rRNA gene sequencing (Fig. 2c). Compared with that in the control group, apoptosis was enriched in the Cis group, while QYG reduced the abundance of this pathway. Meanwhile, multiple pathways related to metabolism were disturbed by cisplatin,

**FIG 1** Legend (Continued)

6C. Photomicrographs were captured at a magnification of 200×. Kidney injury score and fibrosis index were semiquantitatively scored based on the criteria we set. Positive stained area of IHC was quantified using ImageJ. (b) Concentration of BUN in serum was detected using commercial kit. (c) mRNA levels of HAVCR1 and LCN2 in kidney were analyzed by qPCR, and relative gene expression was expressed as relative fold change compared to Cis group. (d) Urine output volume of mice was measured for 12 h before sacrifice. (e) Relative body weight of mice in each group was recorded throughout the experiment. (f) Total protein expression of TGF-$\beta$1, Smad2/3, PARP, and p53 in kidney. GAPDH (glyceraldehyde-3-phosphate dehydrogenase) was used as a loading control. (g) mRNA levels of TNF-$\alpha$ and IL-6 in renal tissue. Data are presented as mean ± SEM ($n = 8$). *, $P < 0.05$; **, $P < 0.01$; and ***, $P < 0.001$.

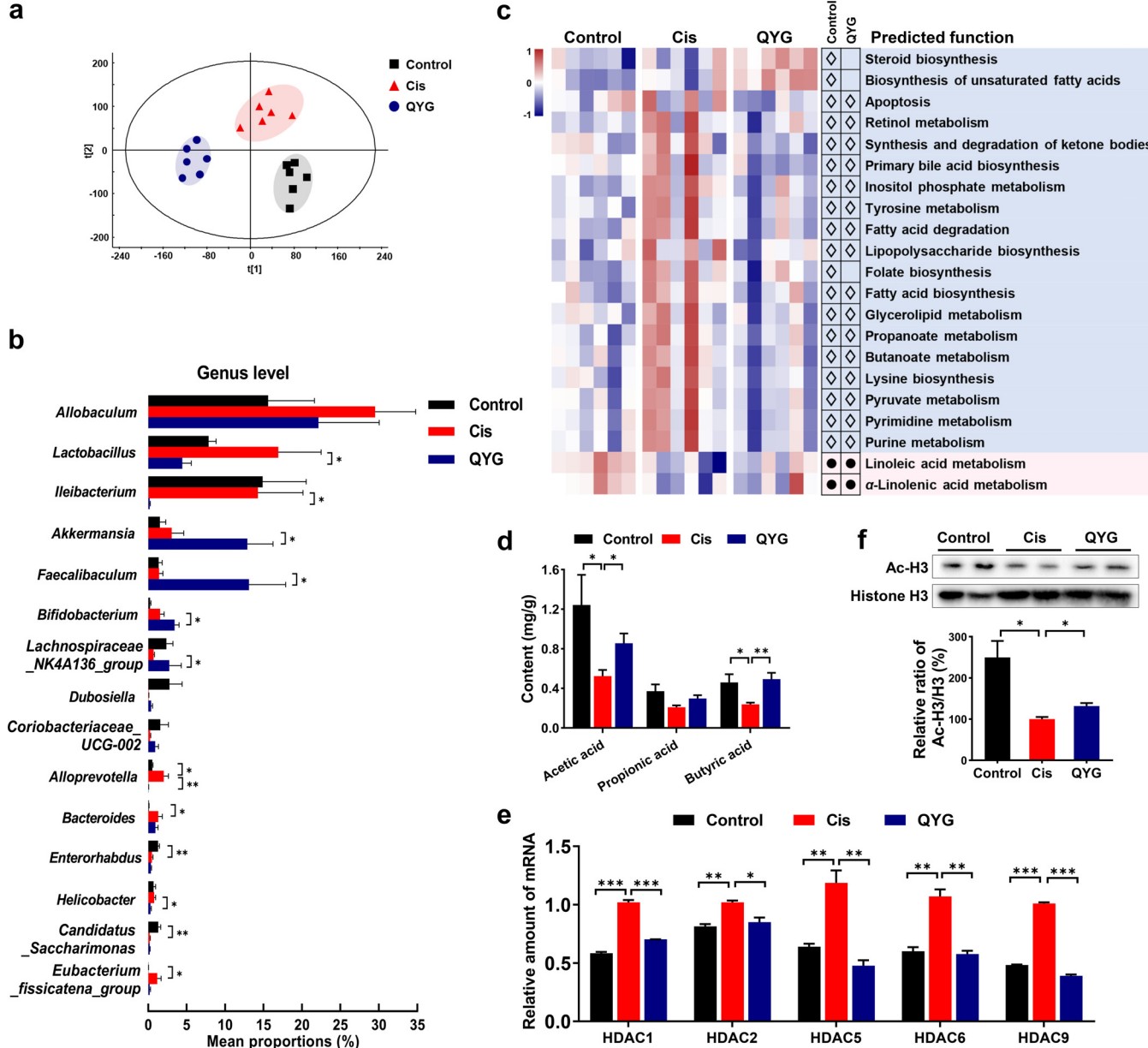

**FIG 2** QYG alleviated cisplatin-induced gut dysbiosis and alteration of SCFAs levels. (a) PLS-DA analysis of gut microbiota composition based on OTUs. (b) Bacterial genera significantly different between groups. (c) Heatmap of functional profiles of microbial communities in control, Cis, and QYG groups. White diamonds represent less abundance in control and QYG groups than in the Cis group, and black dots represent more abundance in control and QYG groups. (d) Contents of acetic acid, propionic acid, and butyric acid in fecal samples were analyzed by GC-MS. (e) mRNA levels of HDACs in kidney. (f) Protein expression of Ac-H3 and histone H3, and data are expressed as relative ratio of Ac-H3/H3 where Cis group was set as 100%. Data are presented as mean ± SEM ($n = 6$). *, $P < 0.05$; **, $P < 0.01$; and ***, $P < 0.001$.

including retinol metabolism, tyrosine metabolism, purine metabolism, and linoleic acid metabolism, while all these changes were reversed after QYG treatment. In addition, increased abundance of lipopolysaccharide biosynthesis was predicted in the Cis group compared with that in the control group, which may lead to the overproduction of lipopolysaccharides to exacerbate renal dysfunction (27). QYG treatment was predicted to alleviate cisplatin-induced abnormal lipopolysaccharide biosynthesis. The above results suggested that QYG alleviated microbiota dysbiosis induced by cisplatin.

**QYG increased levels of acetic acid and butyric acid and suppressed expression/activity of HDACs.** Given the proliferative effects of QYG on putative SCFAs-producing bacteria, SCFAs levels were assessed. Administration of cisplatin reduced the levels of acetic acid, propionic acid, and butyric acid (Fig. 2d), while QYG treatment significantly

increased the levels of acetic acid ($P < 0.05$) and butyric acid ($P < 0.01$). SCFAs are natural inhibitors of HDACs. The results of quantitative PCR (qPCR)-based analysis showed that QYG treatment markedly suppressed cisplatin-induced increase of gene expression of HDAC1, HDAC2, HDAC5, HDAC6, and HDAC9 in kidney undergoing AKI (Fig. 2e). In addition, acetyl-histone H3 (Ac-H3) and histone H3 proteins in renal tissue were detected by immunoblotting. The results showed that acetylation of histone H3 was significantly decreased in Cis group, while QYG treatment increased acetylation of histone H3 (Fig. 2f), which proved that histone deacetylation was suppressed in kidney after inhibition of HDAC expression by QYG.

**QYG protected against cisplatin-induced metabolic disorder.** Untargeted metabolomics analyses of serum, urine, and renal tissue were performed to assess metabolic alternations in response to the gut microbiota remodeled by QYG. Distinct clustering of metabolites was apparent in control, Cis, and QYG groups (Fig. 3a). By performing orthogonal partial least-squares discrimination analysis, we identified 76 metabolites responsible for group discrimination between Cis and control groups, of which 46 were reduced and 30 were increased in the cisplatin group (Fig. S2). QYG treatment regulated 54 of the above metabolites, of which 33 were upregulated and 21 were downregulated compared with those in the Cis group. The pathway analysis revealed that seven metabolic pathways, including linoleic acid metabolism, sphingolipid metabolism, purine metabolism, biotin metabolism, tryptophan metabolism, tyrosine metabolism, and retinol metabolism, were found to be most relevant to the potential biomarkers (Fig. 3b). Pretreatment of QYG remarkably modulated these cisplatin-induced abnormal pathways (excluding linoleic acid metabolism and retinol metabolism) ($P < 0.05$), and the corresponding metabolites (Fig. 3c), which were cer(d18:1/18:0), IDP, biotin, serotonin, and norepinephrine. As a marker of kidney dysfunction, the metabolite creatinine was found to accumulate in serum samples of the Cis group, and QYG treatment exhibited a significant decrease of serum creatinine (Fig. 3c). In addition, two uremic toxins (indoxyl sulfate and *p*-cresyl sulfate), generated from colonic bacterial fermentation, were significantly elevated in serum of the Cis group, whereas they were downregulated by QYG treatment.

**QYG-regulated fecal bacteria correlated with the altered metabolites, the metabolites correlated with biomarkers of AKI, and indicators of fibrosis and inflammation.** Spearman's correlation analysis was used to investigate the correlations between gut microbiota and SCFAs or uremic toxins (Fig. 3d). Acetic acid and butyric acid were positively correlated with QYG-enriched bacterial genera *Akkermansia*, *Faecalibaculum*, *Bifidobacterium*, *Lachnospiraceae_NK4A136_group*, and *Coriobacteriaceae_UCG-002* and negatively correlated with QYG-decreased bacterial genera *Allobaculum*, *Lactobacillus*, *Ileibacterium*, *Alloprevotella*, *Bacteroides*, and *Eubacterium_fissicatena_group*, whereas indoxyl sulfate and *p*-cresyl sulfate exhibited the opposite correlation with the above-mentioned bacteria compared to that of the SCFAs.

The correlations between metabolites and biomarkers of AKI or indicators of fibrosis and inflammation were also evaluated (Fig. 3e). Acetic acid and butyric acid displayed a strong negative correlation with biomarker of AKI, including kidney injury score, BUN, HAVCR1, LCN2, and creatinine, and indicators of fibrosis and inflammation, which are fibrosis index, F4/80, and Ly-6G/Ly-6C, yet showed a notable positive correlation with urine output and relative body weight of mice. Reciprocally, indoxyl sulfate and *p*-cresyl sulfate displayed a remarkable positive correlation with BUN, F4/80, Ly-6G/Ly-6C, and TNF-$\alpha$ yet showed a strong negative correlation with urine output and relative body weight. Indoxyl sulfate also positively correlated with kidney injury score, HAVCR1, LCN2, and fibrosis index.

**The renal protective effects of QYG were diminished by antibiotics.** To investigate whether the renal protective effects of QYG are gut microbiota dependent, QYG-pretreated AKI mice were subjected to antibiotic cocktail (ABX) treatment (Fig. 4a). Consistent with the above effects on cisplatin-induced AKI, QYG treatment greatly reduced renal tubular injury, fibrosis, and renal dysfunction (Fig. 4b to d). However, when the intestinal microbiota was depleted by antibiotics, the renal protective effects

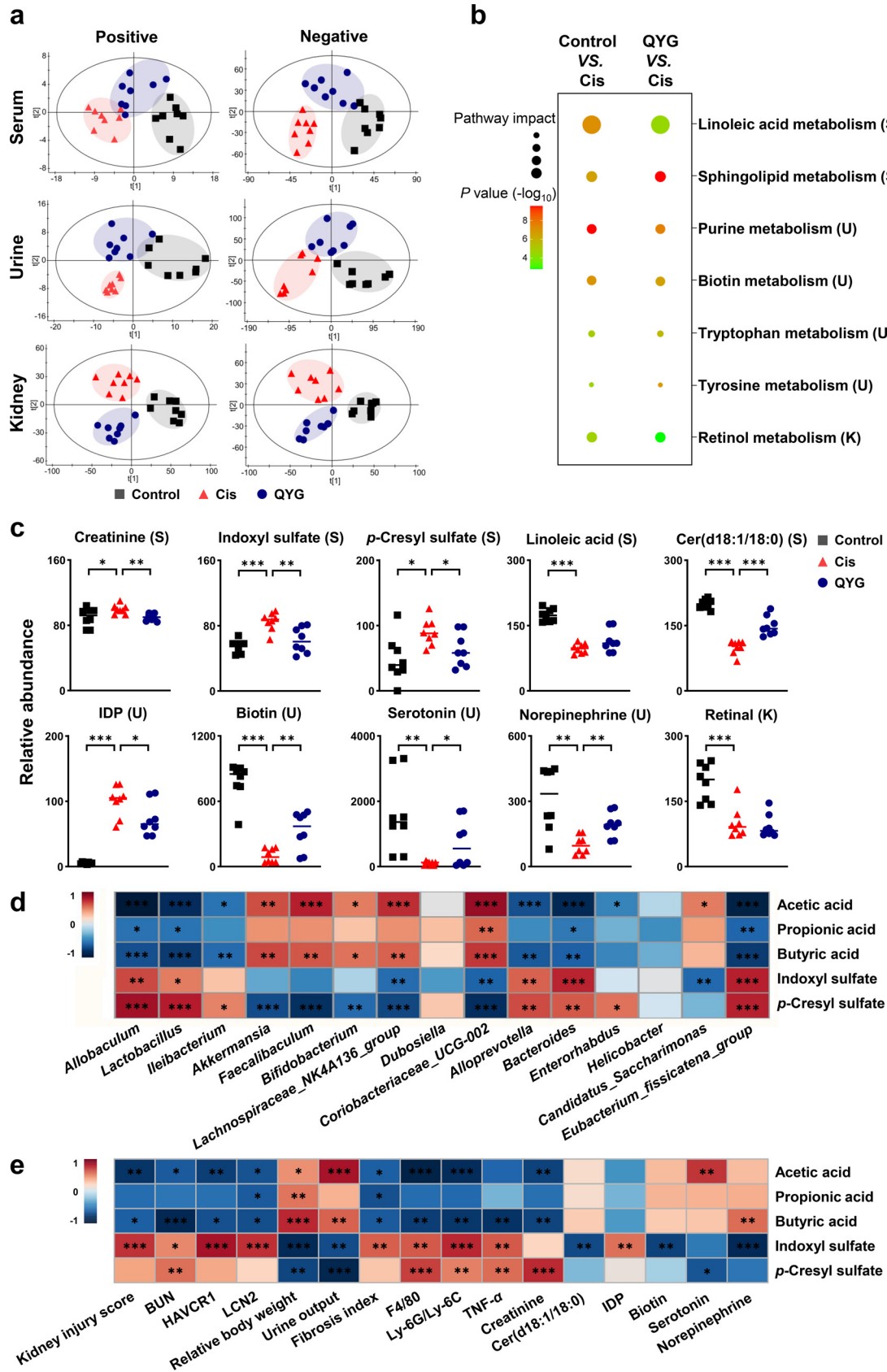

**FIG 3** QYG protected against cisplatin-induced metabolic disorder. (a) PLS-DA score plots of metabolic profiling of serum, urine, and renal tissue in positive and negative ion modes. (b) Disturbed metabolic pathways of control versus Cis and QYG

of QYG were diminished. Obvious necrosis, vacuole formation, and tubular dilation were found in the ABX plus QYG group (Fig. 4b). Meanwhile, ABX plus QYG treatment did not attenuate the tubulointerstitial fibrosis (Fig. 4c), and no significant difference was found in fibrosis index between the ABX plus Cis group and the ABX plus QYG group. Consistently, decrease in the concentration of BUN by QYG treatment was blocked by antibiotic intervention (Fig. 4d). We also found that levels of SCFAs in fecal samples had dropped drastically or even were undetectable when gut microbiota was suppressed (Fig. 4e). These findings indicated that QYG protected against cisplatin-induced AKI in a gut microbiota-dependent manner.

**The renal protective effects of QYG were transferable by FMT.** To further illustrate the renal protective effects of QYG mediated by gut microbiota, fecal microbiota from QYG-treated donor mice were transferred to cisplatin-administered recipient mice (Fig. 5a). After 17 days of colonization, QYG receivers (QYG→Cis group) showed significant regulation of various tubular injuries and tubulointerstitial fibrosis compared with Cis receivers (Cis→Cis group) (Fig. 5b). Renal function of mice in QYG→Cis group was improved, which was evident from decreased level of BUN (Fig. 5c) and gene expression of HAVCR1 and LCN2 (Fig. 5d). Effects of QYG to inhibit the potential mediators of cisplatin-induced AKI were also found in recipient mice. First, the recruitment of inflammatory cells F4/80 and Ly-6G/Ly-6C was diminished (Fig. 5b), and the expression of inflammatory cytokines TNF-$\alpha$ and IL-6 showed the same trend as the QYG group (Fig. 5e). Then, the protein levels of $\alpha$-SMA, TGF-$\beta$1, and Smad2/3 in TGF-$\beta$ fibrosis pathway and apoptosis-related proteins, including PARP and p53, all significantly decreased in the QYG→Cis group compared with those in the Cis→Cis group (Fig. 5b and f).

Horizontal fecal transfer from QYG-treated donor mice demonstrated modulatory effects on SCFAs levels similar to those observed in QYG-treated mice (Fig. 5g). Histone deacetylation was suppressed in kidney of mice from the QYG→Cis group based on the results of decreased gene expression of HDACs and increased protein expression of Ac-H3 (Fig. 5h and i). The metabolic protective effects of QYG were also transferred to recipients, as the QYG→Cis group displayed metabolic profiles different from those of the Cis→Cis group in serum, urine, and renal metabolomics (Fig. S3), and the altered biomarkers, including cer(d18:1/18:0), IDP, biotin, and norepinephrine, were significantly ameliorated (Fig. 5j). Decreased abundance of creatinine and uremic toxins, including indoxyl sulfate and *p*-cresyl sulfate, was detected in QYG receivers compared with that in Cis receivers (Fig. 5j), which was similar to the results shown in the QYG-treated mice. Taken together, these results demonstrated that the nephroprotective effects of QYG were transferable by FMT.

## DISCUSSION

Current research suggests that the gut microbiota plays an important role in the process leading to the onset and progression of renal diseases (8, 28). It has been revealed that gut dysbiosis is closely linked to AKI, and modification of bacterial composition and function might provide treatment options for AKI (29). According to recent clinical studies, *Lactobacillus*, *Alloprevotella*, and *Bacteroides* were prevalent in patients with kidney injury and positively correlated with biomarkers of AKI, whereas probiotics, including *Akkermansia*, *Faecalibaculum*, and *Bifidobacterium*, were enriched in healthy controls (30, 31). In addition, *Allobaculum* has been reported to cause renal dysfunction in diabetic nephropathy C57BL/6 mouse models (32). We found that QYG increased the taxonomic diversity of gut microbiota, which is crucial for conferring re-

**FIG 3** Legend (Continued)

versus Cis groups. The larger the dot, the greater the pathway impact; the darker the color, and the greater the significance. (c) Relative abundance of key metabolites from disturbed metabolic pathways in different groups was expressed as relative increase, where the Cis group was set at 100%. (d) Heatmap summarizing the Spearman's correlation between QYG-regulated fecal bacteria and altered metabolites. (e) Correlations between metabolites and biomarkers of AKI or indicators of fibrosis and inflammation. Red represents positive correlation, and blue indicates negative correlation. Data are presented as mean ± SEM ($n = 8$). *, $P < 0.05$, **, $P < 0.01$; and ***, $P < 0.001$. S, serum; U, urine; K, kidney.

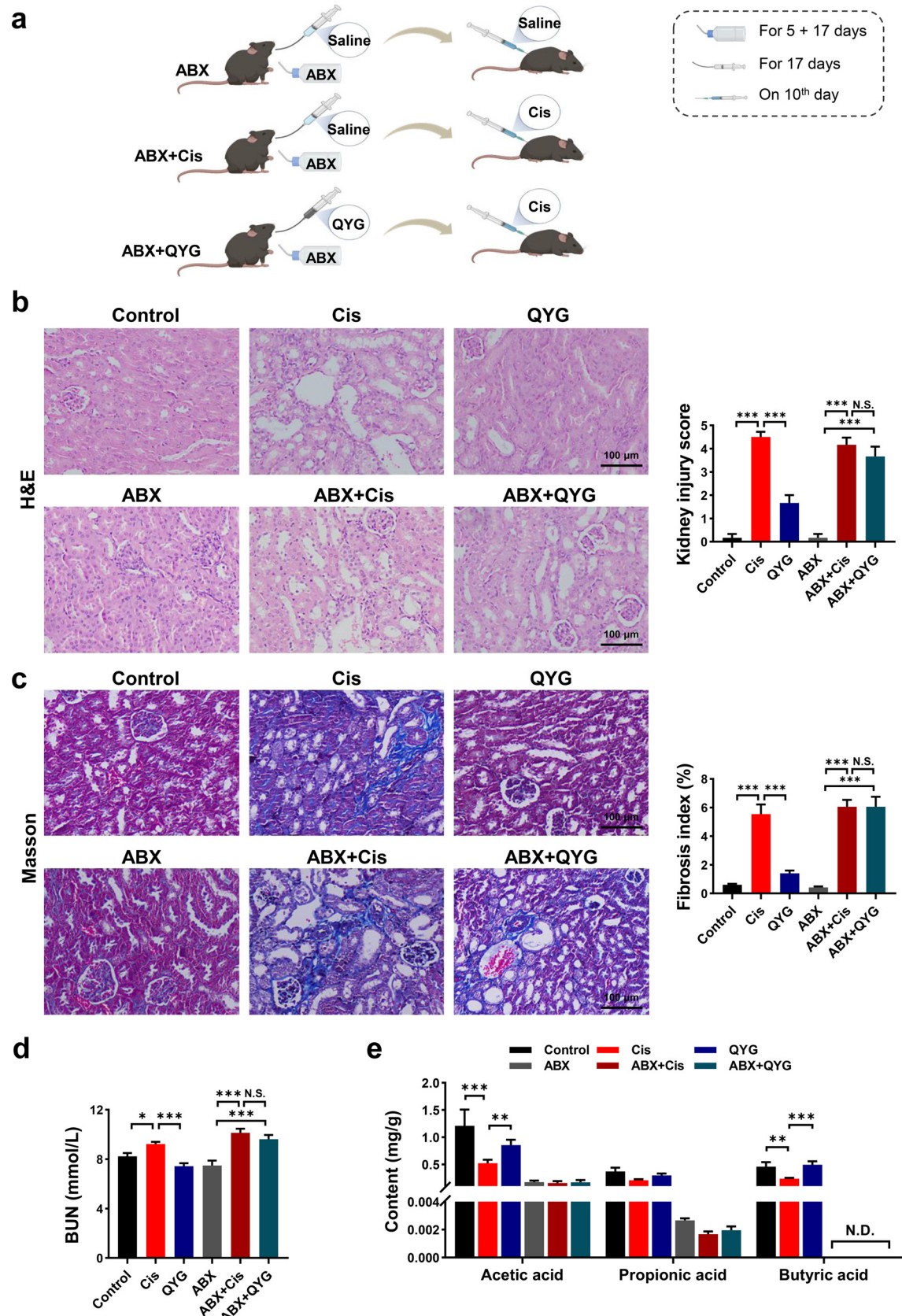

**FIG 4** Renal protective effects of QYG were inhibited by antibiotics. (a) Scheme of antibiotic treatment experiment. Kidney obtained from treated mice was examined with H&E staining (b) and Masson's trichrome staining (c). Photomicrographs were

silence to maintain gut homeostasis (33). In the composition of gut flora, QYG treatment presented reversal effects on the abnormal proliferation of *Allobaculum*, *Lactobacillus*, *Alloprevotella*, and *Bacteroides* induced by cisplatin. Apart from that, QYG treatment also significantly enriched bacterial genera *Akkermansia*, *Faecalibaculum*, and *Bifidobacterium*. Hence, nephroprotective effects of QYG may be related to its modulation of gut microbiota to a healthier profile.

Gut microbiota-derived metabolites are key mediators between the microbiota and the kidney (34). SCFAs, the products of bacterial fermentation from dietary fibers, are recognized as signaling molecules which could affect the renal physiology or even ameliorate kidney injury by acting as HDAC inhibitors (35–38). For example, acetic acid was shown to ameliorate sepsis-induced AKI via attenuation of HDAC activity (36). Treatment with butyric acid can suppress diabetes-induced renal damage through inhibiting HDAC expression and activity (39). On the contrary, uremic toxins like indoxyl sulfate and *p*-cresyl sulfate are potentially deleterious metabolites of gut bacteria, which would contribute to the progression of AKI (34, 40). It has been documented that the levels of indoxyl sulfate and *p*-cresyl sulfate were correlated with severity of AKI patients (41). QYG-enriched probiotics, including *Akkermansia*, *Faecalibaculum*, and *Bifidobacterium*, were all reported to promote the production of SCFAs (42–44). Moreover, the latest study also revealed the ability of *Akkermansia* to regulate tryptophan metabolism, which would contribute to reduced secretion of uremic toxins (45). QYG-regulated bacterial genus *Bacteroides* was reported to promote the production of indoxyl sulfate through tryptophan metabolism (46). Based on metabolomics analysis, renal protective effects of QYG were accompanied by upregulation of acetic acid and butyric acid levels and downregulation of indoxyl sulfate and *p*-cresyl sulfate levels. We further demonstrated that QYG decreased expression and activity of HDAC in renal tissue, suggesting that the increased acetic acid and butyric acid may exert their functions by acting as HDAC inhibitors to alleviate nephrotoxicity. Indoxyl sulfate is synthetized from tryptophan metabolism, while *p*-cresyl sulfate is derived from tyrosine fermentation (34). Intriguingly, QYG was found to regulate the tryptophan metabolism and tyrosine metabolism influenced by cisplatin, which would contribute to reduced secretion of uremic toxins. Collectively, we speculated that the nephroprotective effects of QYG may be related to modulating gut microbiota, then increasing the production of SCFAs, thus suppressing histone deacetylase expression and activity, and reducing the accumulation of uremic toxins.

Fibrosis, inflammation, and apoptosis have been found to contribute to the pathogenesis of cisplatin-induced AKI (47, 48). By acting as HDAC inhibitors, SCFAs have been shown to have pharmacologic activities, including antifibrotic, anti-inflammatory, and anti-apoptotic effects in kidney disease (18). Acetic acid treatment was reported to ameliorate acute and chronic kidney injury in a mouse model through reducing fibrosis and inflammation (38). Butyric acid can suppress high glucose-induced apoptosis in kidney tubular epithelial cells and relieve renal damage and apoptosis in mice (39). Uremic toxins like indoxyl sulfate and *p*-cresyl sulfate would accumulate in renal tissue and stimulate fibrosis, inflammation, and apoptosis, thereby leading to the development of kidney injury (48). QYG treatment was found to attenuate cisplatin-induced AKI by exerting antifibrotic, anti-inflammatory, and anti-apoptotic effects. Spearman's correlation analysis showed that SCFAs or uremic toxins were significantly correlated with the indicators of fibrosis and inflammation. QYG-regulated fecal bacteria were also correlated with these altered metabolites. Taken together, by modulation of gut microbiota, QYG could increase the production of SCFAs, thus suppressing HDAC expression and activity, and reduce the accumulation of uremic toxins, thereby contributing to its nephroprotective effects via restraining fibrosis, inflammation, and apoptosis.

**FIG 4** Legend (Continued)
captured at a magnification of 200×. Kidney injury score and fibrosis index were semiquantitatively scored based on the criteria we set. (d) Concentration of BUN in serum was detected using commercial kit. (e) Levels of SCFAs in fecal samples were analyzed by GC-MS. N.D., not detected. Data are presented as mean ± SEM ($n = 6$). *, $P < 0.05$; **, $P < 0.01$; and ***, $P < 0.001$. N.S., no significance.

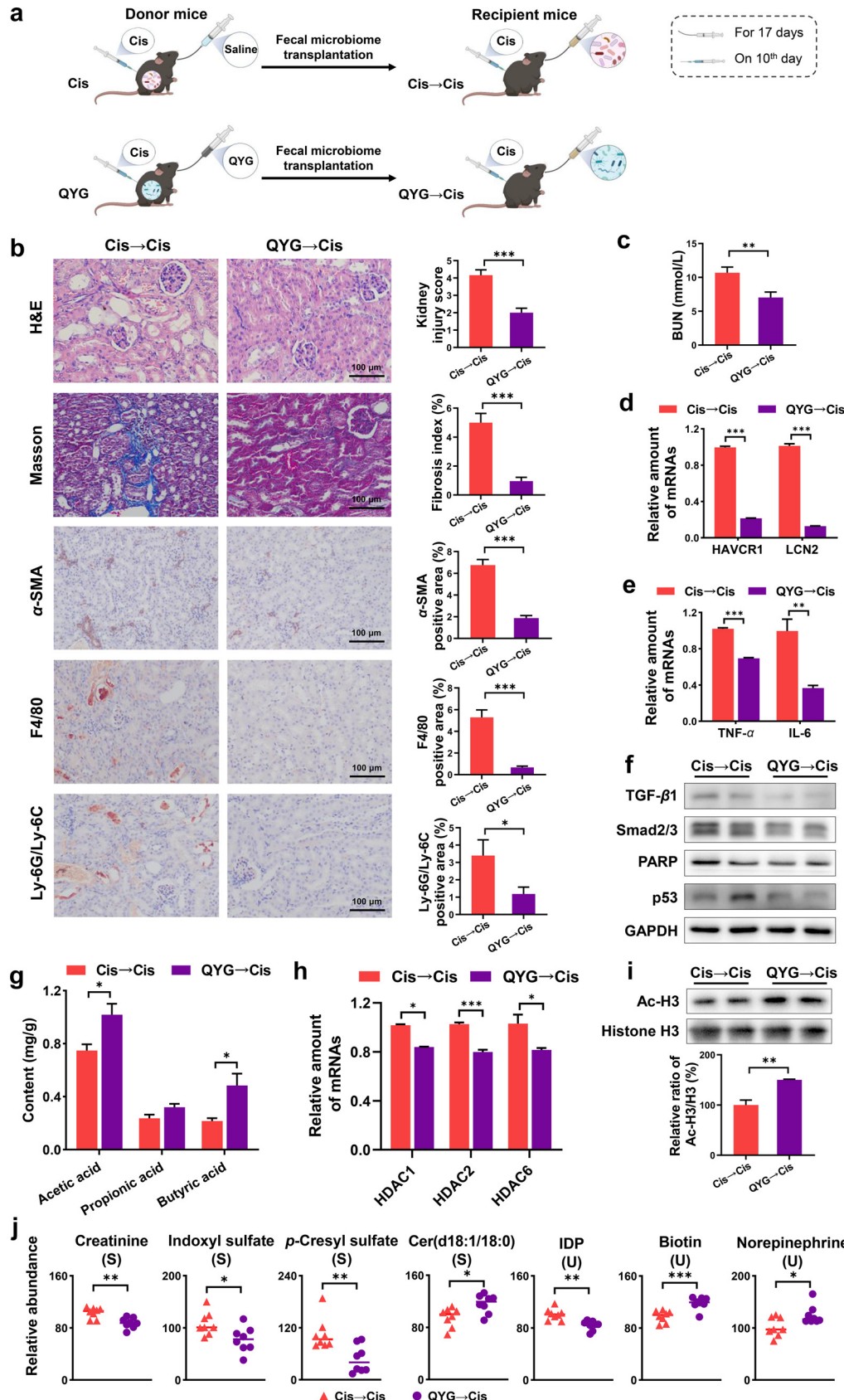

**FIG 5** The renal protective effects of QYG were transferable by FMT. (a) Scheme of FMT experiment. (b) Kidney obtained from treated mice was examined with H&E staining, Masson's trichrome staining, and IHC against α-SMA,

Antibiotics-induced gut microbiota depletion and FMT have been widely used to elucidate gut microbiota-dependent mechanisms of bioactivities (24, 49). In the present study, the renal protective effects of QYG were diminished by antibiotics, and the ability of QYG to alleviate fibrosis and increase SCFA production was also inhibited. Moreover, the renal protective effects of QYG were transferable by FMT. Antifibrotic, anti-inflammatory, and anti-apoptotic effects of QYG, as well as its regulatory effects of SCFAs and uremic toxins, were all recapitulated by FMT. All these results suggested that gut microbiota mediates the nephroprotective effects of QYG against cisplatin-induced AKI, potentially via enriching bacteria genera *Akkermansia*, *Faecalibaculum*, *Bifidobacterium*, and *Lachnospiraceae_NK4A136_group* to increase the production of SCFAs (acetic acid and butyric acid), thus suppressing HDAC expression and activity, and to reduce the accumulation of uremic toxins (indoxyl sulfate and *p*-cresyl sulfate), thereby alleviating fibrosis, inflammation, and apoptosis in renal tissue. Gut microbiota-involved mechanisms of QYG treatment for cisplatin-induced AKI are illustrated in Fig. 6.

In conclusion, our study demonstrated that gut microbiota mediates the protective effects of traditional Chinese medicine formula Qiong-Yu-Gao against cisplatin-induced acute kidney injury. The outputs of this study would provide scientific basis for future clinical applications of QYG as prebiotics to treat cisplatin-induced acute kidney injury, and gut microbiota may be a promising therapeutic target for chemotherapy-induced nephrotoxicity.

## MATERIALS AND METHODS

**Materials.** Rehmanniae Radix and Ginseng Radix were supplied by Jiangsu Province Hospital on Integration of Chinese and Western Medicine (Nanjing, China). Poria was collected from Guizhou Province, the genuine producing area of Poria in China. All herb samples were authenticated according to the monographs documented in Chinese Pharmacopeia (part I, 2020 version). Cisplatin (H20023461) was purchased from Qilu pharmaceutical Co., Ltd. (Jinan, China). Antibiotics, including ampicillin (A70006), metronidazole (M80002), neomycin (N80003), and vancomycin (V10116), were provided by Beijing Jin Ming biotechnology Co., Ltd. (Beijing, China). TGF-$\beta$1 antibody (ab215715) was purchased from Abcam (Cambridge, UK). Antibodies, including Smad2/3 (8685), $\alpha$-SMA (19245), PARP (9532), p53 (2524), histone H3 (4499), and Ac-H3 (9649), were purchased from Cell Signaling Technology (Beverly, MA, USA). F4/80 (14-4801-82) and Ly-6G/Ly-6C (14-5931-82) monoclonal antibodies were from Invitrogen (Carlsbad, CA, USA), and GAPDH (glyceraldehyde-3-phosphate dehydrogenase; 60004-1) was from Proteintech (Rosemont, IL, USA). Reference compounds, including acetic acid (A116165), propionic acid (P110443), butyric acid (B110439), and 2-methylvaleric acid (M117868), were purchased from Shanghai Aladdin Biochemical Technology Co., Ltd. (Shanghai, China). All other reagents used were of at least analytical purity.

**Sample preparation.** QYG was prepared according to the method described in our previous study (22, 25). Briefly, Rehmanniae Radix, Poria, and Ginseng Radix were mixed in a weight ratio of 7:2:1, macerated with 8-fold water for 30 min, and refluxed twice for 4 h each. The combined extraction was filtered, and the filtrates were evaporated under vacuum. Chemical standardization of QYG was performed according to the previous work (25, 26). The representative chromatograms and the analysis results of chemical markers were included in the supplemental material (Fig. S4 to S6 and Table S1). The voucher specimens were deposited in Department of Metabolomics, Jiangsu Province Academy of Traditional Chinese.

**Animals and drug administration.** Male C57BL/6 mice (6 weeks old, weight 20 $\pm$ 2 g) were purchased from SPF (Beijing) Biotechnology Co., Ltd. (Beijing, China). The animals were maintained under standard laboratory conditions at the temperature of 22 $\pm$ 2°C, relative humidity of 55% $\pm$ 5%, and 12-h light/dark cycle, with free access to water and standard chow. All the mice were acclimatized for 1 week before initiation of the experiment. Experimental procedures were carried out in strict accordance with the criteria in the Guide for the Care and Use of Laboratory Animals published by the US National Institutes of Health and approved by the Animal Ethics Committee of Jiangsu Province Academy of

**FIG 5** Legend (Continued)
F4/80, and Ly-6G/Ly-6C. Photomicrographs were captured at a magnification of 200×. Kidney injury score and fibrosis index were semiquantitatively scored based on the criteria we set. Positive stained area of IHC was quantified using ImageJ. (c) Concentration of BUN in serum was detected using commercial kit. (d) mRNA levels of HAVCR1 and LCN2 in renal tissue were analyzed by qPCR. (e) mRNA levels of TNF-$\alpha$ and IL-6 in renal tissue. (f) Total protein expression of TGF-$\beta$1, Smad2/3, PARP, and p53 in kidney. GAPDH was used as a loading control. (g) Levels of SCFAs in fecal samples were analyzed by GC-MS. (h) mRNA levels of HDACs in kidney. (i) Protein expression of Ac-H3 and histone H3, and data are expressed as relative ratio of Ac-H3/H3 where Cis group was set as 100%. (j) Relative abundance of disturbed metabolites. Data are presented as mean $\pm$ SEM ($n = 8$). *, $P < 0.05$; **, $P < 0.01$; and ***, $P < 0.001$.

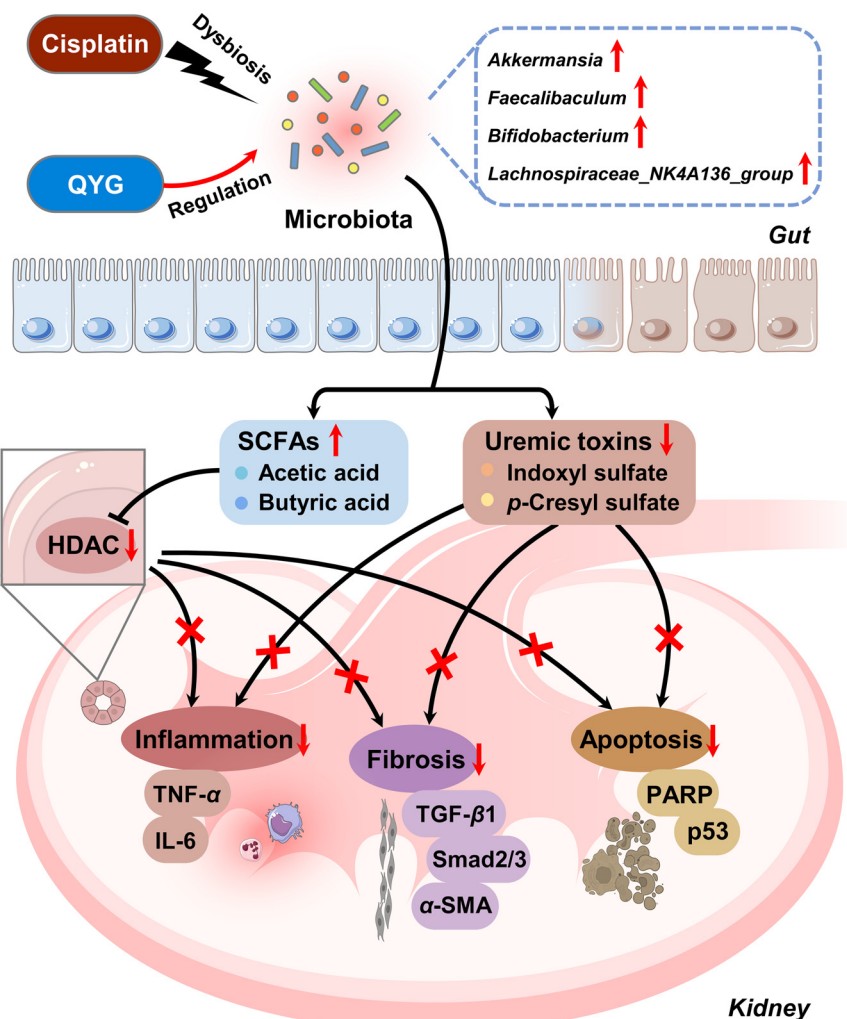

**FIG 6** Schematic illustration of gut microbiota-involved mechanisms in protective effects of QYG against cisplatin-induced AKI. QYG treatment can regulate cisplatin-induced gut dysbiosis, especially enriching *Akkermansia*, *Faecalibaculum*, *Bifidobacterium*, and *Lachnospiraceae_NK4A136_group*. The modulation of gut microbiota enhances the production of SCFAs (acetic acid and butyric acid), thus suppressing HDAC expression and activity, and reduces the accumulation of uremic toxins (indoxyl sulfate and *p*-cresyl sulfate), thereby alleviating fibrosis, inflammation, and apoptosis in renal tissue. Further mechanistic study suggests that antifibrotic effects are mediated through suppressing the activation of TGF-$\beta$1/Smad signaling and that anti-inflammatory effects are mediated by reducing the recruitment of macrophage, neutrophil, and proinflammatory molecules, including TNF-$\alpha$ and IL-6 and anti-apoptotic effects, through inhibiting the activity PARP and p53. Taken together, QYG can attenuate cisplatin-induced acute kidney injury through regulation of gut microbiota.

Traditional Chinese Medicine (IACUC, 201801130).

The induction of the cisplatin-induced AKI model was performed as previously reported by us with some modification (22). Briefly, mice were randomized into three groups (*n* = 12): control, Cis, and QYG. Mice in the QYG group were orally given prepared QYG at a dose of 7 g/kg once daily for 17 consecutive days. Mice in control and Cis groups were administered with equal volumes of saline. Lyophilized cisplatin powder was dissolved in saline and intraperitoneally injected to mice of Cis and QYG groups at a dose of 10 mg/kg on day 10 to establish a cisplatin-induced AKI model. All the mice were sacrificed and dissected on day 17 to collect serum, kidney, and fresh fecal samples.

**Antibiotic treatment and FMT.** Antibiotics were administered *ad libitum* in the drinking water from 5 days before QYG treatment. The broad-spectrum ABX contained 1 g/L of ampicillin, metronidazole, and neomycin and 0.5 g/L of vancomycin (50) and was replaced with freshly prepared ABX once every second day. Other treatment was in accordance with the protocol described in "Animals and drug administration" (Fig. 4A).

FMT experiment was performed based on the established protocol (Fig. 5A). In brief, Cis and QYG groups were the donor mice. The recipient mice were daily treated with fresh transplant material from either Cis or QYG group for 17 consecutive days. Stool was collected daily from donor mice and pooled in sterile tubes. After homogenizing in 10-fold saline, the suspensions were centrifuged at 4,000 × *g* for

5 min. Bacteria-enriched supernatants were collected and treated to recipient mice via oral gavage at 200 $\mu$L per mouse immediately to prevent changes in bacterial composition (49, 51). Mice in recipient groups followed the same method described in "Animals and drug administration" to induce AKI and sacrificed on day 17 to collect serum, kidney, and fresh fecal samples.

**Histological analysis of kidney.** Renal tissue injury was assessed by H&E staining and Masson's trichrome staining. Histopathological examination of the stained tissue was carried out under a light microscope (OLYMPUS CKX41, Tokyo, Japan), and high-power field (200×) photomicrographs were captured. Semiquantification of H&E staining results was performed by histopathological scoring of cortical tubular necrosis. The scoring criteria are set as follows: 0, no injury; 1, minimal injury with less than 10% of cells exhibiting necrosis; 2, mild injury with 10 to 20% cells; 3, moderate injury with 20 to 30% cells; 4, marked injury with 30 to 40% cells; 5, severe injury with more than 40% cells (52). At least 6 representative fields for each slide were scored in a blinded manner by a pathologist. Fibrosis index was used to quantitatively measure the tubulointerstitial fibrosis on Masson's trichrome-stained sections, which was expressed as the percentage of blue collagenous stain in red cellular stain (53). Meanwhile, in paraffin-embedded renal sections, infiltration of macrophages and neutrophils was evaluated by IHC using antibodies against the macrophage marker F4/80 and the neutrophil Ly-6G/Ly-6C marker, respectively. $\alpha$-SMA in TGF-$\beta$/Smad signaling was also analyzed by IHC. The total number of positively stained cells in each group was calculated using ImageJ v. 1.46 software (NIH, USA).

**Renal function determination.** BUN in serum samples was measured using a commercial urea assay kit (ab83362, Abcam) in strict accordance to the manufacturer's standard protocol. For the evaluation of urine output and metabolomics, 12 h of urine samples of each mouse were collected using metabolic cage. Body weight was monitored every day, and relative body weight was calculated as an index of overall well-being.

**Gene expression analysis by qPCR.** The gene expression levels of HAVCR1, LCN2, TNF-$\alpha$, IL-6, and HDAC 1, 2, 5, 6, and 9 were quantified by real-time PCR analysis. Briefly, total RNA was isolated from renal tissue using the FastPure cell/tissue total RNA isolation kit (Vazyme, Nanjing, China) according to the manufacturer's instructions. Isolated RNA was reverse-transcribed to cDNA using HiScript II Q RT SuperMix for qPCR (Vazyme Biotech Co., Ltd.). After that, the qPCR analysis was carried out using ChamQ SYBR color qPCR master mix (Vazyme Biotech Co., Ltd.) on AB StepOnePlus real-time PCR system (Applied Biosystems, Foster City, CA, USA). Sequences of primers used for qPCR are listed in Table S2. Gene expression levels were calculated using the $2^{-\Delta\Delta Ct}$ algorithm after being normalized to the internal control GAPDH and expressed as fold change relative to Cis group.

**Western blotting.** The renal tissue was lysed in RIPA lysis buffer (Beyotime, Shanghai, China) and centrifuged to separate the supernatant containing total protein. Protein was then separated on 12% SDS-PAGE and transferred to polyvinylidene difluoride (PVDF) membranes. After being blocked with 5% skimmed milk in Tris-buffered saline with Tween 20 (TBST) buffer, the membranes were incubated with primary antibodies against TGF-$\beta$1, Smad2/3, PARP, p53, Ac-H3, histone H3, or GAPDH. The membranes were further incubated with horseradish peroxidase (HRP)-conjugated secondary antibodies, visualized using enhanced chemiluminescence (ECL) reagents, and imaged with the Tanon 5260 chemiluminescent imaging system (Tanon, Shanghai, China).

**16S rRNA gene sequencing and analysis.** Fresh fecal samples were collected and stored at $-80°$C until analysis. Extraction of microbial DNA in feces was performed using a QIAamp fast DNA stool minikit (Qiagen, CA, USA). Concentration and purity of the DNA extracted were determined by NanoDrop 2000 UV-vis spectrophotometer (Thermo Scientific, Wilmington, USA), and DNA quality was examined by 1% agarose gel electrophoresis. The V3-V4 regions of the bacteria's 16S rRNA gene were amplified by PCR with the primer pairs 338F (5'-ACTCCTACGGGAGGCAGCAG-3') and 806R (5'-GGACTACHVGGGTWT CTAAT-3'). The resulted PCR products were further purified and quantified using the AxyPrep DNA gel extraction kit (Axygen Biosciences, Union City, USA) and QuantiFluor-ST (Promega, Madison, USA), respectively. The purified amplicons were pooled in equimolar, and sequencing was performed using paired-end configuration on an Illumina MiSeq system (Illumina, San Diego, USA).

The raw data were analyzed on the free online platform of Majorbio Cloud Platform (https://cloud .majorbio.com). The reads were checked with Chimera and assigned to OTU with a 97% similarity threshold. The taxonomy of acquired OTUs was analyzed by RDP Classifier algorithm (http://rdp.cme.msu.edu) against the Silva (SSU123) 16S rRNA database with confidence threshold of 70%. Shannon index and Invsimpson index were calculated using MOTHUR (version v.1.30.1). PICRUSt was applied to predict the potential functional capabilities of the microbiota detected.

**GC-MS-based SCFA analysis.** Reference solutions of acetic acid, propionic acid, and butyric acid were prepared from 0.1 to 500 $\mu$g/mL with pure water (2-methylvaleric acid was used as internal standard and prepared at 100 $\mu$g/mL). Feces were blended in water by homogenizer and centrifuged to separate the supernatant. Internal standard (20 $\mu$L), 50% HCl (50 $\mu$L), and diethyl ether (570 $\mu$L) were added to supernatant fluid or reference solutions (500 $\mu$L). After vortex blending for 1 min, the mixture was centrifuged, and the upper ether solution was taken for GC-MS analysis. Analysis was performed on Aglient 7890B gas chromatograph coupled with an Aglient 7000D MS detector (Agilent, Santa Clara, CA, USA) with an Agilent J&W DB-FFAP column (15 m by 0.25 mm by 0.25 $\mu$m). The mass spectral data were collected in multiple reaction monitoring mode with the characteristic ions of SCFAs obtained using references. Based on acquired data and standard curves generated from references, the SCFAs in fecal samples were determined quantitatively.

**UPLC-QTOF-MS/MS-based metabolomics analysis.** Serum and urine samples used for metabolomics analysis were deproteinized by adding four volumes of ice-cold methanol, and renal tissue was homogenized in 5-fold acetonitrile. All samples were subsequently centrifuged, and the supernatants were

collected and vacuum dried. The dried residue was dissolved in a mixture of water and methanol (1:1, vol/vol) and subjected to metabolomics analysis by UPLC-QTOF-MS/MS.

Liquid chromatographic separation for processed serum, urine, and kidney samples was performed on a Waters ACQUITY UPLC system (Waters, MA, USA) equipped with an ACQUITY UPLC HSS T3 column (2.1 by 100 mm, 1.8 $\mu$m). A Waters QTOF Synapt G2 mass spectrometer (Waters MS Technologies, Manchester, UK) was used for mass detection. Data processing methods were the same as those in our previous study (54). Briefly, peak finding, alignment, filtering, and normalization of raw data were performed using Masslynx v4.1 and Progenesis QI v2.0 (Waters Corporation, Milford, USA). The resultant data matrices were introduced to Ezinfo 2.0 software for partial least-squares discrimination analysis (PLS-DA) to obtain the profile of all samples. Then, variables responsible for group discrimination between control and Cis groups were screened out as potential biomarkers of cisplatin-induced injury. The potential biomarkers were identified based on their UPLC-QTOF-MS/MS data by comparing with the metabolites collected in online databases. The online analytical tool MetaboAnalyst (http://www.metaboanalyst.ca) was used to performed pathway analysis of the identified potential biomarkers.

**Statistical analysis.** All data are presented as mean $\pm$ standard error of mean (SEM). Statistical analyses were performed using GraphPad Prism (GraphPad Software, San Diego, USA). Differences between groups were assessed using unpaired two-tailed Student's $t$ test. The group differences were considered significant with $P$ values less than 0.05. Correlation coefficients were determined using Spearman's correlation analysis.

**Data availability.** The 16S rRNA gene sequencing data have been deposited in the NCBI Sequence Read Archive (SRA) database (https://www.ncbi.nlm.nih.gov/sra) under accession code PRJNA809823.

## SUPPLEMENTAL MATERIAL

Supplemental material is available online only.
**SUPPLEMENTAL FILE 1**, PDF file, 1.8 MB.

## ACKNOWLEDGMENTS

This work was financially supported by the National Natural Science Foundation of China (grant number 81603262), the Natural Science Foundation of Jiangsu Province (grant number BK20161081), the Science and Technology Development Program of Traditional Chinese Medicine of Jiangsu Province (grant number YB2020021), the Project Program of Jiangsu Province Academy of Traditional Chinese Medicine (grant number BM2018024-2019001), and the Postgraduate Research and Practice Innovation Program of Jiangsu Province (grant number KYCX20_1516).

S.-L.L. and F.L. conceived the study. Y.-T.Z., F.L., and J.-H.Z. conducted the experiments. J.Z., J-D.X., S.-S.Z., C.-Y.W., and W.Z. performed sample collection and data analysis. Y.-T.Z., F.L., and S.-L.L. wrote the manuscript and prepared figures. Q.M., H.S., and Y.-Q.Z. helped perform the analysis with constructive guidance. All authors read and approved the final manuscript.

We declare no conflicts of interest.

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
