## [Reviewer comments · Microbiology Spectrum]

Microbiology Spectrum

Gut microbiota mediates the protective effects of traditional Chinese medicine formula Qiong-Yu-Gao against cisplatin-induced acute kidney injury

Ye-Ting Zou, Jing Zhou, Jin-Hao Zhu, Cheng-Yin Wu, Hong Shen, Wei Zhang, Shan-Shan Zhou, Jin-Di Xu, Qian Mao, Ye-Qing Zhang, Fang Long, and Song-Lin Li

Corresponding Author(s): Song-Lin Li, Jiangsu Province Academy of Traditional Chinese Medicine

Review Timeline:

Submission Date:

February 28, 2022

Accepted:

April 12, 2022

Editor: Zhenjiang Xu

Reviewer(s): Disclosure of reviewer identity is with reference to reviewer comments included in decision letter(s). The following individuals involved in review of your submission have agreed to reveal their identity: zhihui zhou (Reviewer #2)

Transaction Report:

DOI: <https://doi.org/10.1128/spectrum.00759-22>

April 12, 2022

Prof. Song-Lin Li
Jiangsu Province Academy of Traditional Chinese Medicine
Department of Metabolomics
Nanjing
China

Re: Spectrum00759-22 (Gut microbiota mediates the protective effects of traditional Chinese medicine formula Qiong-Yu-Gao against cisplatin-induced acute kidney injury)

Dear Prof. Song-Lin Li:

Your manuscript has been accepted, and I am forwarding it to the ASM Journals Department for publication. You will be notified when your proofs are ready to be viewed.

Sincerely,

Zhenjiang Xu
Editor, Microbiology Spectrum

Journals Department
Supplemental file 1: Accept